# Herbicide Resistance: Managing Weeds in a Changing World

Rita Ofosu [1,*], Evans Duah Agyemang [1], Adrienn Márton [1], György Pásztor [1], János Taller [2] and Gabriella Kazinczi [1]

1 Institute of Plant Protection, Georgikon Campus, Hungarian University of Agriculture and Life Sciences, H-8360 Keszthely, Hungary; evansagyemang932@gmail.com (E.D.A.); adriennm2@gmail.com (A.M.); pasztor.gyorgy@uni-mate.hu (G.P.); pacseszakne.kazinczi.gabriella@uni-mate.hu (G.K.)

2 Institute of Genetics and Biotechnology, Georgikon Campus, Hungarian University of Agriculture and Life Sciences, H-8360 Keszthely, Hungary; taller.janos@uni-mate.hu

* Correspondence: rita.of13@gmail.com; Tel.: +36-70-6583704

**Abstract:** Over the years, several agricultural interventions and technologies have contributed immensely towards intensifying food production globally. The introduction of herbicides provided a revolutionary tool for managing the difficult task of weed control contributing significantly towards global food security and human survival. However, in recent times, the successes achieved with chemical weed control have taken a turn, threatening the very existence we have tried to protect. The side effects of conventional farming, particularly the increasing cases of herbicide resistance agricultural weeds, is quite alarming. Global calls for sustainable weed management approaches to be used in food production is mounting. This paper provides detailed information on the molecular biological background of herbicide resistant weed biotypes and highlights the alternative, non-chemical weed management methods which can be used to prevent the development and spreading of herbicide-resistant weeds.

**Keywords:** weed management; resistant weeds; herbicide resistance; glyphosate





## 1. Introduction

Mankind has an incessant problem. A problem if not carefully managed in time could lead to catastrophic results, which could threaten global food security and the livelihoods of billions of people across the globe. This problem is the increasing resistance of weeds to herbicides used in weed management strategies [1–3]. The persistent use of herbicides in food production has led to the increased evolution of many weed species [4–6]. The 'Green Revolution' has become too expensive for biodiversity and is actually putting food production at risk, threatening the very existence mankind has tried to protect for all these years [7–9]. Global population growth is expected to reach about 8.5 billion people in 2030 and 9.7 billion by the year 2050 according to the United Nations Department of Economic and Social Affairs [10]. As pressure increases for more food production to safeguard and sustain global food security for the increasing population, over-reliance on chemical products have become a norm in cultivation in many areas across the globe. The daunting task of curbing global hunger demands a high intensification of production, and over the past couple of decades, industrial agriculture has relied heavily on herbicides for the production of food and animal feed, especially with the development of genetically engineered herbicide tolerant crops [11]. There has been an exponential increase in herbicide use, so much that herbicides account for the largest of pesticides used globally. Intensive agriculture is vital to global food security; however, in most cases, it is highly dependent on chemical products such as pesticides [12]. Highly industrialized food producing areas of The Americas, Australia and China rely on large amounts of herbicides during plant production to manage weeds, and are among the areas with the most recorded cases of herbicide resistance [5]. The global pesticide market size reached a value of nearly $84.5 billion in

2019, out of which herbicide market share accounted for 51.9%, according to a report by The Business Research Company [13]. However, despite the significant contribution herbicides have made towards production by providing a consistent, efficient, rapid and economic approach to weed management, the development of herbicide resistance in many weed species is generating global concerns [14]. Undoubtedly, weeds can be detrimental to crop production due to their ability to outcompete cultivated crops for valuable resources such as nutrients, moisture, light and space. Weeds also serve as reservoirs or hosts of plant pathogens capable of causing diseases to cultivated plants. For example, lamb's quarters (*Chenopodium album*) serves as host for plant virus *Tomato spotted wilt virus* and jimson weed *(Datura stramonium)* serves as host for *Potato virus X* and *Tomato mosaic virus* [15–18]. Livestock production is not spared the damaging effects of weeds. Weed species such as hemlock (*Conium maculatum*) and common ragwort (*Jacobaea vulgaris*) contain toxic alkaloids that cause birth defects, jaundice, or in the worst cases, death when consumed by livestock. Other weed species such as spear thistle (*Cirsium vulgare*) reduce wool quality when it gets on the fleece of sheep during grazing, and stinking chamomile (*Anthemis chamomile*) taint meat and milk flavour when consumed by livestock during grazing on pastures [19]. The potential of weeds to evolve, epigenetic capacity, hybridization, herbicide resistance, herbicides tolerance, cropping systems vulnerability, co-evolution of weeds with human management and the ability of weeds to ride the climate change storm anthropogenic activities have caused weeds to survive management strategies [14]. The introduction of synthetic herbicides in agriculture in the 1940s marked a turning point in food production allowing for extensive cropping of agricultural produce to meet the demands of the food. In spite of the increased production volumes recorded over the years, revelations and situations arising from the past couple of years since the introduction of these chemicals are undermining the success of herbicide use [20]. Over-reliance and extensive use of herbicides in our farming systems worldwide is leading to the development of herbicide resistant weeds. This paper seeks to discuss herbicide resistance in modern agriculture, highlighting potential weed management strategies being used to minimize the rate of evolution of weeds and the devastating effect herbicide resistance is causing globally.

## 2. Herbicide Resistance in Industrialized Farming Systems

Intensive monoculture has encouraged the use of herbicides in weed control. This production system has encouraged the use of mainly the same groups of herbicides for weed management during production. This specialized cultivation of a single crop as observed in the commercial production of crops including wheat, soybean, corn, canola and cotton on large scale usually relies heavily on herbicides for weed management. Since the discovery of herbicide resistance in common groundsel (*Senecio vulgaris*) to triazine herbicides in 1968, there have been several recorded cases of resistances of weeds to several chemical groups across the globe [21,22]. Weeds have evolved resistance to 21 out of 31 known herbicide action sites and to 165 different herbicides. Currently, herbicide resistance has been reported in 96 crops in 72 countries, and 513 unique cases of weed resistance involving 267 species have been reported globally [23]. Currently, herbicide sustainability in agriculture is being greatly challenged. Often times, it is recommended that chemical methods be the last resort in pest management; however, due to the efficiency and efficacy of chemical products against targets, they have become persistently and extensively used globally. The adoption and utilization of synthetic herbicides in food production over the past century has triggered the evolution of resistance among various weeds species to a lot of herbicide chemical classes making weed management more difficult [24–27].

The United States of America (USA), Europe, Australia, Canada, Brazil and China—where farming systems are industrialized—lead in the numbers of recorded cases of resistant weeds [23]. However, unlike in the USA, Australia, Canada and Brazil—where cases of herbicide resistance are linked to the cultivation of biotech crops—herbicide resistance in Europe is associated with low diversification of chemical products used in controlling weeds. Biotech crops such as glyphosate-resistant crops have allowed for the single

use of glyphosate, resulting in increasing cases of glyphosate resistant weeds. Countries in Europe have recorded several cases of weed biotypes resistant to acetolactate synthase (ALS) inhibiting herbicides, acetyl-CoA carboxylase (ACCase) inhibiting herbicides, 5-enolpyruvylshikimate-3-phosphate (EPSP) synthase inhibiting herbicides, photosystem II (PS II) inhibiting herbicides and fatty acid synthase inhibitors. Some weed species identified and recorded to have developed herbicide resistance include *Conyza* species resistant to flazasulfuron, iodosulfuron, and penoxsulam in vineyards, olive and citrus farms. *Lolium* species have also been recorded to have evolved resistance to glyphosate. *Avena* species, *Bromus* specie, *Digitaria sanguinalis*, *Panicum dichotomiflorum* and *Echinochloa crus-galli* resistance to ALS-inhibiting herbicides have been recorded in some countries in Europe [23,28]. Some individual active ingredients of herbicides have recorded more cases of resistant weed species across the globe. Most of the areas with the recorded cases of resistance to these active ingredients are usually associated with the monoculture farm systems as shown in Table 1.

**Table 1.** Number of resistant weed species to individual active ingredients (top 15) sourced from the International Herbicide-Resistant Weed Database [23].

| Active Herbicide(AI) | Mode of Action of AI | Number of Weed Species | Areas with Reported Cases of Weed Resistance | Field Situations |
|---|---|---|---|---|
| Atrazine | Inhibitor of photosynthesis at photosystem II | 66 | North America, South America, Oceania (Australia and New Zealand), Europe, Asia and South Africa | Wheat, canola, corn, sugar beets, cotton, soybean, industrial sites, cotton, roadsides, winter wheat, tomatoes, sorghum, cropland, vegetables, pastures, forests, orchards, nurseries, blueberries, carrots, lupins, turf, asparagus |
| Glyphosate | Inhibitors of EPSP synthase | 57 | North America, South America, Oceania (Australia and New Zealand), Europe, Asia and South Africa | Wheat, cereals, canola, spring barley, lentils, peas, golf courses, grapes, winter barley, corn, fallow, chickpeas, clover, irrigation channels, cotton, soybean, spring barley, industrial sites, roadsides, winter wheat, alfalfa, sorghum, pasture seed, beans, hazel nut, rice, coffee, orchard, oil palm, nurseries, lime, almonds, pumpkin, squash, turf, apples, canola |
| Tribenuron-methyl | Inhibitors of acetolactate synthase | 45 | North America, New Zealand, Europe, Asia and South Africa | Wheat, canola, corn, sunflower, lentils, winter barley, alfalfa, cropland, fallow, beans, cereals, winter wheat, grapes, roadsides, rapeseed, durum wheat, peas, chickpeas, spring barley, pastures |
| Imazethapyr | Inhibitors of acetolactate synthase | 44 | North America, South America, Australia, Europe and Asia | Wheat, canola, corn, sunflower, soybean, industrial sites, cotton, railways, onions, lentils, alfalfa, cropland, fallow, rice, cereals, sorghum, vegetables, peas, forests, chickpeas, spring barley, lettuce, cabbage, peanut |
| Imazamox | Inhibitors of acetolactate synthase | 40 | North America, South America, Australia, Europe, Asia and South Africa | Wheat, sunflower, canola, spring barley, lentils, chickpeas, rice, rapeseed, alfalfa, orchard, cropland, grapes, sugar beets, sorghum, kentucky bluegrass, tomatoes, winter wheat, corn, cotton, soybean, nurseries, cereal |
| Metsulfuron-methyl | Inhibitors of acetolactate synthase | 39 | North America, South America, Australia, Europe, Asia (Iran and Malaysia) and South Africa | Wheat, sunflower, canola, spring barley, lentils, chickpeas, rice, cropland, grapes, winter wheat, winter barley, faba beans, lupins, peas, roadside, pastures, grass seed, industrial sites, oil palm, turf, corn, cotton, soybean, cereals |

**Table 1.** *Cont.*

| Active Herbicide(AI) | Mode of Action of AI | Number of Weed Species | Areas with Reported Cases of Weed Resistance | Field Situations |
|---|---|---|---|---|
| Chlorsulfuron | Inhibitors of acetolactate synthase | 38 | North America, Europe, Asia and Oceania (Australia and New Zealand) and South Africa. | Wheat, canola, spring barley, lentils, chickpeas, fallow, cropland, winter wheat, winter barley, faba beans, lupins, peas, roadside, pasture seed, industrial sites, forests, corn, cotton, soybean, sorghum, cereals |
| Iodosulfuron-methyl-sodium | Inhibitors of acetolactate synthase | 38 | North America, South America, Oceania (Australia and New Zealand), Europe and South Africa | Wheat, sunflower, canola, spring barley, lupin, golf courses, grapes, winter wheat, corn, cotton, soybean, spring wheat, durum wheat, pasture seed |
| Fenoxaprop-P-ethyl | Inhibitors of fat synthesis acetyl coA carboxylase inhibitors (ACCase inhibitors) | 33 | North America, South America, Oceania (Australia and New Zealand), Europe, Africa and Asia | Wheat, sunflower, canola, lentils, peas, sugar beets, chickpeas, winter wheat, winter pulses, flax, cropland, spring barley, winter wheat, corn, cotton, soybean, spring wheat, beans, spring wheat, Faba beans, lupins |
| Paraquat | Inhibitor of photosynthesis at photosystem I | 31 | North America, South America, Oceania, Europe, Africa and Asia | Wheat, pears, pasture seeds, chestnut, peaches, sugarcane, vegetables, corn, cotton, soybean, spring wheat, beans, oil palm, tea, almonds, coffee, rubber, sweet potato, roadside, blueberries, hobs, taro, orchard, mulberry, alfalfa |
| Simazine | Inhibitor of photosynthesis at photosystem II | 31 | North America (USA), Australia, Europe and Asia | Cropland, roadside, nurseries, vegetables, canola, spring barley, forests, industrial sites, golf courses, grapes, winter wheat, corn, sugar beets, potatoes, blueberries, spring wheat, durum wheat, pasture seed |
| Bensulfuron-methyl | Inhibitors of acetolactate synthase | 29 | Australia, China South Korea, Japa, Malaysia, Portugal, Italy, Chile, Spain, Turkey, USA | Rice, gulf courses |
| Thifensulfuron-methyl | Inhibitors of acetolactate synthase | 29 | North America, Europe, Asia, New Zealand and South Africa | Wheat, cereals, canola, spring barley, lentils, peas, golf courses, grapes, winter barley, cropland, corn, cotton, soybean, spring barley, industrial sites, roadsides, winter wheat, alfalfa, tomatoes, sorghum, pastures |
| Mesosulfuron-methyl | Inhibitors of acetolactate synthase | 26 | USA, Chile, Australia, Europe, Asia and South Africa | Wheat, cereals, chickpeas, rapeseed, watermelon, triticale, carrots, durum wheat, rice, pastures, sugar beets, peas, grapes, winter barley, cropland, corn, cotton, winter wheat, spring barley |
| 2-4-D | Synthetic Auxin (Plant cell growth disruptor) | 25 | North America, South America, Oceania (Australia and New Zealand), Europe and Asia | Wheat, cereals, winter barley, spring barley, cropland, corn, pastures, roadsides, soybean, oats, turf, sugarcane, rice |

*Molecular Background of Herbicide Resistance*

Most herbicides block the function of an essential plant enzyme by binding to it. A change in a single amino acid at the binding region may lead to the plant's survival, even with a cost in fitness, as in resistance to photosystem II inhibitors. How these mutations develop is unclear, but evidence was found that point mutations endowing resistance to herbicides can be present in weed populations as part of their genetic variation. The selection pressure of the applied herbicides would facilitate the proliferation of such biotypes [3]. In the herbicide-resistant weed registry [23], out of the 267 weed species which are registered for any herbicide resistance, in 57 species (>20%), multiple resistant cases are reported, i.e., resistance to different herbicide groups in the same biotype. These plants

withstand various toxins, each with another target. Two sites of action (SoA) resistances were reported for 53 species, three SoA for 17, four SoA for 10, five SoA for six and six and seven SoA resistances are reported for one-one species for a *Lolium rigidum* and a *Poa annua* biotype, respectively. In some species, several biotypes were identified with different numbers and compositions of SoA's. It should also be noted that in about one-third of the weed species, resistance to single herbicides evolved in different biotypes, which in some cases was quite a high number. For example, for *Lolium rigidum* or *Poa annua*, resistance was found against nine and ten herbicide groups, respectively. Considering that resistance to different herbicide groups can accumulate in a single biotype, the consequences of an exclusively herbicides-based weed control strategy can be foreseen. Most herbicide target genes reside on chromosomes, and mutant alleles of these can be transmitted by pollen or the egg cell to the next generation. The accumulation of mutant genes and genetic mechanisms conveying resistance to different herbicide groups is an evolutionary response to herbicides' intense and continuous selection pressure. Even against non-target site herbicides, such as synthetic auxins, resistance evolved at least to 11 herbicides. The mechanism of action of synthetic auxins (HRAC Group 4) is quite complex. These induce intensive plant growth, leading finally to senescence and death of the affected tissues. Against these broad-spectrum herbicides, such as 2, 4-D or dicamba, resistance was already registered in 62 weed species. Currently, 31 multiple resistance cases involving resistance to a synthetic auxin and one or more other herbicide group were reported [23]. It can be concluded that combining synthetic auxin resistance and another target site herbicide resistance in cultivated plants, as it happens in some newly developed genetically modified (GM) cultivars, can potentially lead to the evolution and spread of multiple resistant weeds. The development of such weeds reveals the evolutionary flexibility of plants. Less sensitive weeds to these herbicides that survive herbicide spraying grow and spread. Since resistant genes are able to spread by hybridization between related species, there is the possibility of the accumulation of resistance genes in different biotypes [2].

One of the problems in efficiently fighting herbicide resistance and keeping the effectivity of herbicides is the lack of timely information. Farmers are aware of the problem only when the decreased efficiency of the used chemical is already evident as weed populations survive. Currently, no easily applicable detection methods exist to survey herbicide resistance on an arable field. Consequently, pollens and seeds can transfer the characteristics to closer and more distant areas. To sustain the effectiveness of herbicides, detecting methods for evolving resistance should be widely administered.

An interesting question is whether a resistant biotype would persist in the long run when that herbicide is not used further in that area. This question could be answered with field trials using non-herbicide-resistant crop cultivars. It is hypothesized that the normal type will dominate the weed population for biotypes where herbicide survival accompanies a cost in fitness after stopping the selection pressure. A similar hypothesis cannot be drawn in cases with no fitness cost. To answer such questions, a detailed understanding at the molecular level is required to elaborate reliable monitoring methods of herbicide resistance that could serve as a cornerstone in deciding the appropriate weed control method. Many publications are available about the types of resistances, their evolution, function and possible detection approaches. Here, a brief overview of the relevant molecular aspects and mechanisms of herbicide resistance is given. Herbicide resistance of weeds is categorized as target site or non-target site resistance. In target site resistance (TSR), the binding of the inhibitory agent to an essential plant enzyme is disturbed by amino acid substitution(s). By this, the enzyme escapes the blocking effect of the chemical and can execute its catalytic function. Non-target site resistance (NTSR) is often complex, involving genes of large gene families, and its mechanism includes reduced absorption or translocation, and increased sequestration or metabolic degradation [25,29]. In the case of ACCase, ALS, PDS or PPO inhibiting herbicides, the successful evolution of TSR is that the herbicide doesn't block the active site of the enzyme where conservative amino acids reside. Instead, it binds in that region but mainly to less conserved amino acids, which can be substituted with others,

resulting in a functional isoform of the original enzyme to which the herbicide cannot bind [30]. A further factor in the evolution of TSR is that the herbicide is not fitting precisely the substrate envelope of the enzyme, and energetically the binding is not beneficial because it is established primarily with amino acid radicals and not with the carbon backbone of the protein. In contrast to those inhibitors, glyphosate and glufosinate compete with the substrate molecule because these fit tightly into the active site of the target enzyme and bind with strong hydrogen bridges to it. Hence, amino acid substitutions in the active site could result in the dysfunction of the enzyme. Although non-synonymous mutations are known for glyphosate resistance, which affect a single codon, these induce a weak resistance to the herbicide. Instead, duplications of the target gene are the common mechanism for glyphosate resistance in weeds [31–34].

Diverse and complex mechanisms contribute to NTSR, also involving different gene families in many cases. Due to the complexity, much fewer molecular details are explored for NTSR than for TSR. The absorption of herbicides administered to the plant foliage shows differences attributed mainly to anatomical features such as cuticle thickness, number and structure of trichomes. Reduced herbicide absorption through the plasma membrane is also a mechanism for the altered transport process. Reduced translocation, when the herbicides are transported in lower than necessary amounts to the site of action in the plant, can also be one reason for insufficient phytotoxicity. A type of NTSR is when, although the herbicide can enter the plant because of rapid vacuolar sequestration, it cannot be translocated to the site of action. Nevertheless, reduced absorption is considered just one contributing factor to the NTSR of the plant. In plants, several enzyme families are known to neutralize herbicides. Members of the enzyme superfamily cytochrome P450 monooxygenases are localized in the endoplasmic reticulum, and play a crucial role in the metabolism of endo- and exogenous substances. Their role in the metabolic resistance to herbicides has been proven in numerous weeds. Another type of NTSR is executed by another enzyme superfamily, the glutathione S-Transferases (GST), which detoxify herbicide molecules by conjugating them to glutathione. Glucosyltransferases inactivate herbicides by conjugating glucose molecules to phase I modified herbicide metabolites. Phase I modifications are hydroxylation or demethylation. In monocots resistance to synthetic auxins is connected to glycosylation of the hydroxylated rings of the herbicide [29,35].

Researchers have continuously stressed using herbicides of different sites of action in production to prevent resistance development. Herbicide mixtures in treatment strategies have been reported to increase the efficiency of chemical agents in weed control [36]. Some research has also shown that mixing different modes of action (MOA) of herbicides does not provide permanent solutions in target site resistance since it only delays the evolution of the weeds [37]. Glyphosate has become dominantly used as a non-selective herbicide in controlling a wide spectrum of weeds in these industrialized food producing areas since its introduction in 1974 [38,39]. The advent of genetically engineered glyphosate-tolerant crops in the 1990s equipped the crop production industry with a revolutionary tool for weed management, leading to the intense utilization of glyphosate worldwide [40,41]. Glyphosate-tolerant crops, mainly soybean, cotton, corn and canola have become extensively cropped. The over-reliance on glyphosate in the commercial cultivation of engineered crops such as soybean and corn have triggered TSR in weeds in many areas [2,42]. There have been several reports globally on economically important weed species developing multiple resistance mechanisms and exhibiting multiple resistance across many herbicide classes [43–47].

Besides point mutations at the herbicide binding site of the target gene, other mechanisms can also evolve, which convey resistance against the herbicide. Such a mechanism is the amplification of the target gene. Plants can develop resistance by amplifying the genes that encode the target proteins of herbicides. Through gene duplication multiple copies of the gene can be present in resistant individuals. This increased gene dosage provides more copies of the target protein, diluting the herbicide's effect and allowing the plant to tolerate higher herbicide concentrations [48]. Further, plants can develop resistance also by altering their metabolic pathways to detoxify or degrade herbicides. This can occur

through changes in the activity or abundance of specific enzymes involved in herbicide metabolism. Mutations or changes in the regulation of these enzymes can enhance their ability to break down or modify the herbicide molecules, reducing their toxicity to the plant [49].

Another mechanism of herbicide resistance involves reducing the uptake or translocation of herbicides within the plant. This can occur through mutations that affect the functioning of transport proteins involved in herbicide uptake or movement within plant tissues. By limiting the herbicide's entry into sensitive sites, resistant plants can avoid its detrimental effects [50]. Nevertheless, it is important to note that the specific changes in DNA leading to herbicide resistance can vary depending on the plant species and the herbicide in question. Additionally, resistance mechanisms can be complex and often involve multiple genetic changes working together.

## 3. Herbicide Use in Non-Industrialized Farming Systems

The adoption and use of herbicides have been slow in non-industrialized farming systems as compared to industrialized farming systems. However, in recent times, the use of herbicides is increasing in areas where traditional farming systems previously relied on mechanical weed control methods [51]. One of such areas seeing an increase in the use of herbicides is Africa. The African continent is keen to produce enough food to feed its fast-growing population [52]. Although the adoption of chemicals in agriculture has been slow in Africa, recent evidence suggests a steady increase in the use of herbicides in food production across the continent, with more and more growers resorting to chemical products for weed management since these products give growers a more cost effective and rapid alternative to weeding [53]. Regulations concerning herbicides and herbicide application in many parts of Africa are not properly monitored and regulated. Farming systems in many sub-Saharan countries such as Ghana are largely traditional and informal. Though herbicide use in Africa accounts for about 2 to 4% of global market share of pesticides and can be considered amongst the lowest rate of their usage in the world [54], it is still common to find growers use herbicides in their production. Most of the growers are smallholder farmers who have little or no knowledge of the herbicide mode of action and herbicide resistance. Moreover, there are instances where the precautions and recommendations for herbicide application and usage are not followed, thereby increasing the risk of resistance development. Instances where growers do not read or comprehend information on the labels of chemical products being used is common. As a result, there is the possibility of farmers applying the wrong rates of herbicides. A reduction in the application rate of herbicides increases the ability of weeds to evolve resistance. The situation is compounded by the influx of mislabeled, adulterated and cheap pesticides available on the market for sale as pesticides [55–59]. In a region where food security is already threatened by variable climate and political instability, inadequate infrastructure and insufficient biosecurity measures to monitor and limit the spread of weed species, as well as to prevent the accidental introduction of invasive weeds into the environment, the current situation is extremely worrying and begs for critical attention to be paid to developing appropriate and proper regulations on synthetic herbicides use to prevent catastrophic consequences resulting from herbicide resistance in the region in the near future [54,60]. Chemical products which are banned or severely restricted elsewhere such as organochlorine pesticides (OCPs) and Atrazine, are still being used in some parts of Africa for weed control [61,62]. South Africa has recorded several cases of weed resistance to several herbicides. There have been cases of resistance of *Raphanus raphanistrum*, *Avena fatua*, *Lolium rigidum*, *Phalaris minor*, *Stellaria media*, *Amaranthus palmeri*, *Amaranthus hybridus*, *Conyza bonariensis* and *Plantago lanceolata* to ALS, ACCase, EPSP, PSII and PSI herbicides in South Africa [63]. The most recent record of reported cases of resistant weeds in Africa indicate that South Africa has a record of 15 cases of unique resistant weeds; Egypt with three reported cases, followed by Ethiopia and Kenya with one reported case each [23]. The limited amount of research on herbicide resistance in the region could be attributed to the limited record of data and information on

weed resistance in the region. In spite of the fact that genetically modified crops are not as extensively spread and cultivated in Africa as they are in other parts of the world, the threat of over-reliance on herbicide as the sole weed control method is gradually increasing each year as more and more youth migrate from rural areas to urban areas in search of employment, leaving behind deficits in rural labour demand, as well as increasing the cost of manual labour [63,64]. The aged farmers left in the rural communities are also not able to properly undertake certain traditional farming activities such as hand weeding or hoeing, forcing farmers to rely on a cheaper and faster alternative by using chemical methods in weed control [59]. There is widespread prevalence of low quality, fraudulent glyphosate products being sold at various African markets. Issues pertaining to lack of comprehensive herbicide regulations is common to many countries in developing areas across the globe [65,66]. Since the expiration of Monsanto's patent in 2000, there has been an increase in the production of glyphosate by agrochemical companies across the globe. The importation of glyphosate into African countries over the past couple of years has increased, as more growers continue to rely on herbicides [67]. Research has confirmed and established glyphosate-resistant hairy fleabane (*Conyza bonariensis*) cases in the western and southern Cape regions of South Africa. Currently, South Africa, Sudan, Egypt, Nigeria, Malawi, Ethiopia and Burkina Faso are among the few African countries leading the commercialization of GM crops. Other countries such as a Rwanda, Ghana, Mozambique and Niger have made significant progress in crop research, and have prospects of planting GM crops in the future [68].

## 4. Herbicide Threat to Environment

This dependency on herbicides is beginning to affect the biodiversity that exists in our environment. The direct effect of applying herbicides against non-targets such as bees, butterflies and spiders which are beneficial organisms necessary for pollination and crop protection is sometimes deadly [69]. The toxic effects of atrazine, which is able to persist in soils for a long time, easily leached into ground water or indirectly washed into surface is devastating to aquatic flora and fauna. The contamination causes massive destruction to the different life forms that exist in the aquatic environment. Its effects include a reduction in reproduction and spawning of fish species, and a reduction of algae biomass in streams and rivers close to agricultural lands where herbicides are used [70]. Spray drifts and leachates from target sites during the application of herbicide pose a threat to the surrounding vegetation [71]. Research has found herbicides are being ingested by humans through the consumption of fruits and vegetables with herbicide residues on them. The contamination could be as a result of spray drift, inappropriate herbicide application time or chemical concentration. This may result in damage of the nervous and reproductive systems, developmental abnormalities and organ failure [72].

## 5. Non-Chemical Weed Control Techniques Used in Weed Management

Non-chemical weed control techniques provide alternative approaches to herbicide application in weed management. The techniques involved in non-chemical weed control are useful and capable of managing herbicide resistant weeds, and has been practiced across the globe as a weed control method since the beginning of domestic cultivation. It often comprises preventative, physical and biological approaches to controlling weeds [46,73].

### 5.1. Preventative, Cultural, Physical and Mechanical Weed Control

Preventative and cultural weed control measures are usually aimed at the successful establishment of cultivated plants and boosting crop competitiveness against weeds growing on fields [74]. Several studies have shown that practicing good cultural farming activities is able to reduce weed infestations to tolerable levels. Factors such as tillage, irrigation, appropriate planting time, proper sowing methods, planting density, farm and farm equipment sanitation, cultivar, cropping system, mulching and plant spacing are important to consider when developing weed management strategies for production [75–79]. Growers

in both developed and developing countries use physical and mechanical methods of weed control. Physical weed control methods such as hand weeding or hoeing have been used to manage weeds during cultivation for centuries and continue to be used in some parts of the world for weed. However, the labour intensiveness, time required, and high cost requirement in commercial scale cropping makes it ineffective in controlling weeds. Mechanical weed control has proven successful in controlling weeds. This method employs the use of implements to allow for large tracts of land to be infested with weeds to be cleared. However, it is also faced with limitations with weather, cropping pattern, high cost of operation, type of weed and danger to biodiversity making it not as effective as chemical weed control [80,81].

### 5.2. Biological Weed Control

This form of weed control depends on natural mechanisms such as predation and parasitism by naturally occurring organisms to control weeds [82]. Bioherbicide techniques use allelochemicals, natural byproducts, plant extracts, microorganisms and insects as control mechanisms. In recent times, advances have been made using biological control agents in weed management, with current research focus on the potential evolutionary changes after release [83,84]. The bioherbicides are capable of disrupting photosynthesis, nutrient uptake and other functions necessary for plant survival. There is huge potential for the technique in weed management; however, the adoption and commercialization of bioherbicides in weed management has been slow due to limiting factors such as the environment, formulation, possible toxicity to non-targets and in some cases cost. Although bioherbicides provide a greener and safer alternative to synthetic herbicides, it currently cannot rival the achievements of synthetic herbicide in weed control [85–88]. Bioherbicides play an important role in sustainable weed management as they can be used in combination with other management methods to provide an effective control mechanism. This method of weed control requires appropriate and proper planning to be effectively executed [87].

## 6. Managing Weeds in Modern Farming

The reality is the global world population is constantly growing and food security is a major concern, particularly in developing areas. More food is required to be produced to sustain the masses. Unfortunately, the impact of current climate issues pertaining to drought, floods and changing weather patterns, compounded by the increasing rates of herbicide resistance in weed, seems to be putting limitations on food production and biodiversity [89]. Although there has been significant improvement in food cultivation strategies over the years, weeds are successfully adapting and evolving mechanisms to meet human engineered solutions for weed control. It can be argued that the efficacy of herbicides has been compromised [90,91]. However, is there an alternate brilliant technology which can rival the results herbicides have achieved in weed control today? Are herbicides dispensable in 21st century commercial agriculture? Can there be a compromise between herbicides and weeds as the world struggles to find means of reaching set goals for attaining global food security? Irrespective of the answers we ascribe to these questions, we cannot deny that weeds and herbicides are important to the survival of mankind. Therefore, the need for a critical approach to solving the problem at hand. Modern agriculture needs to be innovative enough to shift from the paradigm of sole dependence on a particular approach in solving weed-related problems. Several studies in the past couple of years since the discovery of resistant weed biotypes to herbicides in the 1950s has continuously emphasized the need for better weed management approaches to limit this increasing evolution [92,93].

### Sustainable Approaches to Weed Management

The problem currently at hand with herbicide resistance demands a holistic approach to weed management that deviates from the single tactic of relying solely on herbicides for weed control to an approach that efficiently and effectively utilizes multiple weed

management tactics in production while ensuring it is eco-friendly and less detrimental to biodiversity [94]. The current state and occurrence of herbicide resistance in the next couple of decades is predicted to worsen, creating even more of a challenge towards sustainable agriculture and the preservation of biodiversity. Studies conducted over the years have shown the importance of integrated weed management (IWM) as a weed management approach. The approaches employed in management strategies should effectively combine monitoring, prevention and control methods to achieve the target. Currently, there is no absolute weed management program; however, there is a high chance of reducing weed evolution and resistance development using diverse management systems which are best suited to conditions present at a particular place [95,96]. The integration of all available control methods (preventative, cultural, mechanical, biological and chemical) is vital to achieving optimum results in weed management. This integration of control methods is crucial in reducing the chemical footprint in weed management which in certain situations has been producing results that have improved production with less negative impact on the environment. A study conducted by Lehnhoff et al. [97] reported on how combinations of integrated grazing and herbicide, specifically grazing with fall-applied rimsulfuron or imazapic, provided better results than treatment with herbicide alone against Cheatgrass (*Bromus tectorum* L.). In the study, the livestock grazed the field in the spring and chemical control was applied in the summer and fall seasons when the target weeds were dominant. The use of a combination of biological (livestock grazing) and chemical control methods as observed in the study demonstrated how the use of multiple stressors at different growth stages can help achieve the desired results of a weed management strategy without relying solely on chemicals, and is one way of reducing the occurrence of herbicide resistant weeds, providing a more sustainable and environmentally friendly approach to weed management. Sustainable weed management (SWM) in recent years has shown promising results in utilizing a variety of strategies to control weeds, and there is increasing support for sustainable management strategies in many parts of the world [98–103]. Global concerns and calls for improved growing methods geared towards organic crop production with little or no reliance on synthetic pesticides to ensure consumer safety have been increasing. These concerns being raised demands for a reduction in the use and risks associated with hazardous pesticides in food production [104]. The European Union (EU) is amongst the areas with the most stringent and comprehensive pesticide legislation in the world. However as highlighted earlier, pesticide regulations and laws differ across the globe. Since herbicides still form a major part of weed control, the concerns being raised are warranted. This is in order to create awareness and facilitate the setting up and adherence to appropriate legislation to prevent possible issues associated with chemical application [54,105]. Cultural weed control methods such as crop rotation, cover cropping and intercropping have been highlighted by studies to be effective approaches to improving crop competition by maximizing light, water and nutrient capture. These cropping systems are effective in suppressing weeds without resorting to herbicides for weed control. A reduction in herbicide use minimizes the development of resistance mechanisms in weeds. Crop rotation can directly disrupt the growing cycle of weeds which helps in reducing infestation levels. Allelopathy is a biological phenomenon of chemical interaction between plants that can be used as a tool in crop rotation, intercropping, cover cropping and mulching to manage weeds. The use of allelopathy as an alternative approach to the use of synthetic herbicides provides an effective and efficient tool for controlling weeds in a sustainable manner in both conventional and organic farming. There is evidence of sorghum-wheat rotation decreasing dry weed biomass due to the accumulation of sorghum allelochemicals (sorgoleone). Other crops such as alfalfa, sunflower, corn and wheat have been found to reduce weed densities when included in a crop rotation cycle due to the presence of allelochemicals. It is, however, important to pay particular attention to the choice of crops in a rotation, since allelochemicals can have adverse effects on cultivated crops in the rotation as well [73,103,106,107].

Cultivar competitiveness has proven to provide a sustainable approach to improving plant vigor against weeds. Some cultivars of wheat have been found in a study to possess competitive ability traits that can improve weed management. Crop-weed competition can be influenced by the type of weed species, time of emergence of the weeds and the density of weeds on the growing field. Weeds may vary in their ability to compete with cultivated crops due to differences in species, despite having the same densities. Perennial weeds have been found to be more competitive than annual weeds due to their early vigour, dense shoot growth and deep roots. The magnitude of yield loss is usually high when weeds emerge before or simultaneously to cultivated crops. This is due to the ability of weeds to establish faster; hence, the crops do not compete well for resources. Growing crop cultivars with traits such as fast germination, large leaf area, high biomass, rapid growth and development can improve crop resilience against weeds on a growing field. As such, the detrimental effect on yield due to weed competition could be highly reduced. This is particularly important for weed management, since herbicide application will be limited [106,108]. In order to achieve a more sustainable, resilient and ecologically friendly approach to weed management, there is also the need for the adoption of diversified cultivation systems that will be less dependent on monoculture and sole dependence on one method of weed control, particularly chemical weed control. With monoculture, as stated earlier, a single type of crop is usually grown on large tracts of land over a long period. In areas where applicable, crop diversification should be practiced to limit the issues that come with monoculture, such as herbicide resistance [109]. Recent developments in the areas of precision agriculture, particularly with site-specific weed management technology combines information systems and sensors to effectively control weeds in an environmentally friendly approach [110–113]. There is increasing popularity of modern advancement in robotics, remote sensing, artificial intelligence and modelling as valuable tools used in determining weed dynamics and estimating infestation levels for weed control in sustainable agriculture [114]. Recent advances in herbicide application technology as observed in precision weed control are vital in reducing reliance on chemical products for weed management in modern agriculture, and reducing the rapid evolution of resistance of weeds. Since the application can detect important changes in the weed population and also apply herbicides accordingly, cases of resistant weeds could be significantly reduced effectively using this technology.

## 7. Conclusions

Managing weeds during production is important in achieving optimum yield. An effective weed management strategy must include cultural, mechanical, biological and chemical methods of weed control. In a situation where herbicides are the principal components of a weed control program, growers need to consider available herbicides, the type of cropping system, application rates, label recommendations, cultivar tolerance to herbicide, soil type and the type of weeds present on growing fields when planning and developing weed management programs. Timing herbicide application is very important and growers should apply pre-emergence and post-emergence herbicides at the right time. It is important to time the application of herbicide to suit the appropriate growth stage of the weed in order to achieve optimum results. Usually, herbicides are applied when weeds are young, since weeds are more susceptible to the effects of the herbicide at this stage. Herbicide rotation is important in preventing the establishment of resistant weeds. Rotating one herbicide group with other herbicide group(s) that control the same weeds growing on a field during a growing season or over a period of years can delay resistance development. However, it is important to consider the mode of action and the site of action of the herbicides used in the rotation to prevent incidences of multiple resistance development. There is a need for the active participation of all stakeholders and government leaders to ensure sustainability by strengthening national regulatory laws to limit the indiscriminate use of herbicides in cultivation, particularly in areas with weak regulatory institutions. In areas where necessary, capacity building should be a priority with focus on improving the knowledge base of growers on herbicide application and

resistance development. Information on the right use of herbicide and alternatives should be made easily accessible to growers to promote sustainable weed management. Weed researchers will need to adopt an interdisciplinary and transdisciplinary research, since herbicide resistance continues to occur at an alarming pace. Solving and managing the problem of herbicide resistance requires the combined efforts of all stakeholders, including growers, ecologists, agronomists, weed scientists, plant breeders and herbicide developers.

**Author Contributions:** Conceptualization, R.O.; resources, data curation, writing and original draft preparation, R.O., J.T., G.P. and A.M.; editing, R.O. and E.D.A.; review and supervision, G.K. and J.T. All authors have read and agreed to the published version of the manuscript.

**Funding:** This study was funded by the Hungarian Government and the European Union, with co-funding of the European Regional Development Fund in the frame of Széchenyi 2020 Programme GINOP-2.3.2-15-2016-00054 project.

**Institutional Review Board Statement:** Not applicable.

**Informed Consent Statement:** Not applicable.

**Data Availability Statement:** Not applicable.

**Acknowledgments:** This study was supported by the Hungarian Government through the Stipendium Hungaricum Scholarship Program, Tempus Public Foundation and the Government of Ghana.

**Conflicts of Interest:** The authors declare no conflict of interest.

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
