# Peer review of "Herbicide Resistance: Managing Weeds in a Changing World"

_agronomy, doi:10.3390/agronomy13061595_

Round 1
Reviewer 1 Report
The review paper was written under the direction of the Agronomy Journal, and fulfills all the criteria for publication. Problems covered by the paper are interesting, with high contribution to the field. Quality of writing it should be commended. Introduction to the problem that was treated in this paper is written clearly and concisely. Well written discussion set out logical conclusions. Literature data are in line with the topic of the paper.
This manuscript will be acceptable after minor revision (corrections to minor error and text editing). In line 267 – for hairy fleabane should be added latin name Erigeron bonariensis.
Author Response
Thank you for very much for your useful and detailed comments. We have addressed the your suggestion as follows:
This manuscript will be acceptable after minor revision (corrections to minor error and text editing). In line 267 – for hairy fleabane should be added Latin name Erigeron bonariensis.
We thank Reviewer 1 for this valuable suggestion.
Line 296, We have modified the section to include the Latin name of hairy fleabane. However instead of Erigeron bonariensis we added Conyza bonariensis as was stated in the literature material we cited.
Please find attached the revised manuscript.

Reviewer 2 Report
The review by Ofosu et al entitled: "Herbicide Resistance: Managing Weeds in a Changing World" is focused on weed management strategies and the obstacle of herbicide resistance. The authors described the important problem of weed evolutionary resistance response to herbicides and summarized several mechanisms of the weed responses. New strategies and future recommendations are summarized as well.
Comments to authors:
The issue described and the questions raised by the authors are most important, however although the authors elaborated on describing the problems, the new strategies are only summarized and need extensive elaboration on the successful management (paragraph 6.1).
Additional comments:
1. Lines 48-50. Regarding the detrimental weed effects, please add weeds as reservoirs of disease causing agents.
2. Line 51. Did you mean damaging effects of weeds or of herbicide treated weeds on livestock? Please correct.
3. Line 68. It is not clear how monocultures are related to encouragement of herbicide use. Is it only statistically related?
4. Lines 84-87. It is not clear how herbicide resistance is associated with cultivation of biotech crops.
5. Line 88. Please explain abbreviations.
6. Line 134-136. This is an important issue: how genetically modified cultivars affect herbicide resistance. Please elaborate and add examples. Lines 205-210 are related to this issue.
7. Line 213. Is this line still related to genetically modified crops?
8. Lines 231-233. Are there precautions on use of herbicides that reduce herbicide resistance development in weeds?
9. Lines 373-376. How do crop rotations and diversified cultivation affect weed control? Line 376 correct to 'competitiveness'.
10. Line 398. How does timing of herbicide application prevent weed resistance?
11. Line 400. Doesn’t the use of 'herbicide rotation' cause weed resistance against multiple herbicides?
Author Response
We thank you for the detailed and useful comments. Please find the attached document for the revised manuscript. We have addressed each of the comments as follows:
The issue described and the questions raised by the authors are most important, however although the authors elaborated on describing the problems, the new strategies are only summarized and need extensive elaboration on the successful management (paragraph 6.1).
We regret that the information on the issue described was inadequate. Accordingly, we have added relevant information to parts of paragraph 6.1. Parts of the section have been rewritten. We hope there is more elaborations on the topic now.
Line 389-395. The approaches employed in management strategies should effectively combine monitoring, prevention and control methods to achieve target. Currently, there is no absolute weed management program however, there is a high chance of reducing weed evolution and resistance development using diverse management systems which are best suited to conditions present at particular place [92, 93]. Integration of control methods (preventative, cultural, biological and chemical) have been produce better results.
Line 398-408. In the study, the livestock grazed the field in the spring and chemical control was applied in the summer and fall seasons when the target weeds were dominant. The use of a combination of biological (livestock grazing) and chemical control methods as observed in the study demonstrated how the use of multiple stressors at different growth stages can help achieve the desired results of a weed management strategy without relying solely on chemicals is one way of reducing the occurrence of herbicide resistant weeds providing a more sustainable and environmentally friendly approach to weed management. Sustainable weed management (SWM) in recent years has shown promising results in utilizing a variety of strategies to control weeds and there is increasing support for sustainable management strategies in many parts of the world [95-100].
Line 433-452. Crop-weed competition can be influenced by the type of weed species, time of emergence of the weeds and the density of weeds on the growing field. Weeds may vary in their ability to compete with cultivated crops due to differences in species despite having the same densities. Perennial weeds have been found to be more competitive than annual weeds due to their early vigour, dense shoot growth and deep roots. The magnitude of yield loss is usually high when weeds emerge before or simultaneously as cultivated crops. This is due to the ability of weeds to establish faster hence the crops do not compete well for resources. Growing crop cultivars with traits such as fast germination, large leaf area, high biomass, rapid growth and development can improve crop resilience against weeds on a growing field. As such, the detrimental effect on yield due to weed competition could be highly reduced. This is particularly important for weed management since herbicide application will be limited [103, 105]. Also, in order to achieve a more sustainable, resilient and ecologically friendly approach to weed management, there is the need for the adoption of diversified cultivation systems that will be less dependent on monoculture and sole dependence on one method of weed control particularly, chemical weed control. With monoculture, as stated earlier, a single type of crop is usually grown on large tracts of land over a long period. In areas where applicable, crop diversification should be practiced to limit the issues that come with monoculture such as herbicide resistance [106].
Line 460-462. Since the application can detect important changes in weed population and also apply herbicides accordingly, cases of resistant weeds could be significantly reduced effectively using this technology.
Lines 48-50. Regarding the detrimental weed effects, please add weeds as reservoirs of disease causing agents.
Line 52-56. Weeds also serve as reservoirs or hosts of plant pathogens capable of causing diseases to cultivated plants.
Line 51. Did you mean damaging effects of weeds or of herbicide treated weeds on livestock? Please correct.
In response to the question asked and suggestion given, the section has been modified to the statement below
Line 56-62. Livestock production is not spared the damaging effects of weeds. Weed species such as hemlock (Conium maculatum) and common ragwort (Jacobaea vulgaris) contain toxic alkaloids that cause birth defects, jaundice, or in worst cases death when consumed by livestock. Other weed species such as spear thistle (Cirsium vulgare) reduces wool quality when it gets on the fleece of sheep during grazing and stinking chamomile (Anthemis chamomile) taint meat and milk flavour when consumed by livestock during grazing on pastures [19].
Line 68. It is not clear how monocultures are related to encouragement of herbicide use. Is it only statistically related?
In response to the question asked, the section has been modified to the statement below
Line 77-81. This production system has encouraged the use of mainly the same groups of herbicides for weed management during production. This specialized cultivation of a single crop as observed in the commercial production of crops including wheat, soybean, corn, canola and cotton on large scale usually relies heavily on herbicides for weed management.
Lines 84-87. It is not clear how herbicide resistance is associated with cultivation of biotech crops.
In response to the statement, the section has been modified to include the statement below
Line 100-101. Biotech crops such as glyphosate-resistant crops have allowed for the single use of glyphosate resulting in increasing cases of glyphosate resistant weeds.
Line 88. Please explain abbreviations.
In response to the suggestion, the section has been modified to explanations of the abbreviations
Line 103-104. ...acetolactate synthase (ALS) inhibiting herbicides, acetyl-CoA carboxylase (ACCase) inhibiting herbicides, 5-enolpyruvylshikimate-3-phosphate (EPSP) synthase inhibiting
Line 134-136. This is an important issue: how genetically modified cultivars affect herbicide resistance. Please elaborate and add examples. Lines 205-210 are related to this issue.
In response to the comments, the section has been modified to include
Line 155-158. Less sensitive weeds to these herbicides that survive herbicide spraying grow and spread. Since resistant genes are able to spread by hybridization between related species there is the possibility of accumulation of resistance genes in different biotypes [2].
Lines 205-210 are related
Line 221-226 tells the situation of the general use of herbicide mixtures in weed management. It tells the development of target site resistance regardless for all kind of herbicides used.
Line 213. Is this line still related to genetically modified crops?
Line 228-234. Yes, the line is related to genetically modified crops. However, this point seeks to highlight glyphosate-tolerant crops and target site resistance in weeds present at growing sites of glyphosate-tolerant crops.
Lines 231-233. Are there precautions on use of herbicides that reduce herbicide resistance development in weeds?
In response to the question, the section has been modified to include
Line 257-260. Instances where growers do not read or comprehend information on the labels of chemical products being used is common. As a result, there is the possibility of farmers applying wrong rates of herbicides. Reduction in the application rate of herbicides increases the ability of weeds to evolve resistance.
Lines 373-376. How do crop rotations and diversified cultivation affect weed control? Line 376 correct to 'competitiveness'.
In response to the question, the section has been modified to include
Line 416-430. These cropping systems are effective in suppressing weeds without resorting to herbicides for weed control. Reduction in herbicide use minimizes the development of resistance mechanisms in weeds. Crop rotation directly can disrupt the growing cycle of weeds Allelopathy a biological phenomenon of chemical interaction between plants that can be used as a tool in crop rotation, intercropping, cover cropping and mulching to manage weeds. The use of allelopathy as an alternative approach to the use of synthetic herbicides provides an effective and efficient tool for controlling weeds in a sustainable manner in both conventional and organic farming. There is evidence of sorghum-wheat rotation decreasing dry weed biomass due to the accumulation of sorghum allelochemicals (sorgoleone). Other crops such as alfalfa, sunflower, corn and wheat have been found to reduce weed densities when included in a crop rotation cycle due to the presence of allelochemicals. It is however important to pay particular attention to choice of crops in a rotation since allelochemicals can have adverse effects on cultivated crops in the rotation as well [70, 100, 103, 104].
Line 431. “Competitiveness” corrected as directed.
Line 398. How does timing of herbicide application prevent weed resistance?
In response to the question asked, the section has been modified
Line 470-475. Timing herbicide application is very important and growers should apply pre-emergence and post-emergence herbicides at the right time. It is important to time the application of herbicide to suit the appropriate growth stage of the weed in order to achieve optimum results. Usually, herbicides are applied when weeds are young since weeds are more susceptible to the effects of the herbicide at this stage.
Line 400. Doesn’t the use of 'herbicide rotation' cause weed resistance against multiple herbicides?
In response to the question asked, the section has been modified
Line 476-480. Rotating one herbicide group with other herbicide group(s) that control the same weeds growing on a field during a growing season or over a period of years can delay resistance development. However, it is important to consider the mode of action and the site of action of the herbicides used in the rotation to prevent incidences of multiple resistance development.
